# Identifying differences in gait adaptability across various speeds using movement synergy analysis

David Ó'Reilly[1,2]*, Peter Federolf[3]

1 Catherine McAuley School of Nursing & Midwifery, Brookfield Campus, University College Cork, Co. Cork, Ireland, 2 Faculty of Biological sciences, School of Biomedical sciences, University of Leeds, Leeds, United Kingdom, 3 Department of Sport Science, University of Innsbruck, Innsbruck, Austria

* David.oreilly166@gmail.com

## Abstract

### Introduction

The aim of this study was to identify movement synergies during normal-walking that can differentiate healthy adults in terms of gait adaptability at various speeds. To this end, the association between movement synergies and lower-limb coordination variability or Deviation Phase (DP) was investigated. This study also investigated the moderating effect of movement synergies on the relationship between DP and the smoothness of arm-swing motion (NJI).

### Method

A principal component analysis of whole-body marker trajectories from normal-walking treadmill trials at 0.8m/s, 1.2m/s and 1.6m/s was undertaken. Both DP and NJI were derived from approx. 8 minutes of perturbed-walking treadmill trials. Principal movement components, $PM_k$, were derived and the RMS of the $2^{nd}$-order differentiation of these $PM_k$ ($PA_kRMS$) were included as independent variables representing the magnitude of neuro-muscular control in each $PM_k$. Each $PA_kRMS$ were input into maximal linear mixed-effects models against DP and (DP x NJI) respectively. A stepwise elimination of terms and comparison of models using Anova identified optimal models for both aims.

### Results

The principal movement related to the push-off mechanism of gait ($PA_4RMS$) was identified as an optimal model and demonstrated a significant negative effect on DP however this effect may differ considerably across walking-speeds. An optimal model for describing the variance in (DP x NJI) included a fixed-effect of $PA_6RMS$ representing Right—Left side weight transfer was identified.

### Interpretation

The hypotheses that individuals who exhibited greater control on specific kinematic synergies would exhibit variations during perturbed walking was substantiated. Supporting

**Citation:** Ó'Reilly D, Federolf P (2021) Identifying differences in gait adaptability across various speeds using movement synergy analysis. PLoS ONE 16(1): e0244582. https://doi.org/10.1371/journal.pone.0244582

**Data Availability Statement:** The current study is a secondary analysis of publically available data (https://peerj.com/articles/918/) which can be freely acquired. Supplementary data attached within the current study provides insight into this

secondary analysis and can be found at the following Github repository: https://github.com/Davidoreilly12/O-Reilly-Federolf-2020.

**Funding:** The authors received no specific funding for this work.

**Competing interests:** The authors have declared that no competing interests exist.

**Abbreviations: CNS**, Central nervous system; **CRP**, Continuous relative phase; **DP**, Deviation phase; **ICC**, Inter-class correlation; **NJI**, Normalised Jerk Index; **PA_k**, Principal accelerations; **PCA**, Principal component analysis; **PM_k**, Principal Movements; **PV_k**, Principal velocities; **RMS**, Root-mean-square.

evidence for the role of movement synergies during the double-support phase of gait in proactively correcting balance was presented as well as the potential for this approach in targeted rehabilitation. The potential influence of leg dominance on gait adaptability was also discussed. Future studies should investigate further the role of walking-speed and leg dominance on movement synergies and look to generalize these findings to patient populations.

## 1.1 Introduction

The stability and adaptability of human movement during quiet stance and locomotion is an area of increasing relevance in the growing aging population. Approximately 30% of older adults will experience a fall leading to extensive personal, social and economic costs [1,2]. A stable gait pattern can be achieved when the effects of small perturbations throughout each stride can be limited, when balance can be recovered successfully from large perturbations and by not encountering perturbations in the environment that exceed the limits of the system [3]. To prevent small perturbations at each stride from accumulating into serious instability, human walking takes on an inverted pendulum like motion that effectively harnesses the biomechanical constraints of the lower-limbs to allow for smooth step-to-step transitions that are both energy efficient and adequately stable [4]. When required, postural control strategies are also introduced by the central nervous system (CNS) to accommodate fall prevention. These may include a rapid change in swing-foot placement trajectory and modulation of stance-foot plantar-flexion at push-off [5,6]. This may also be coupled with a change in arm-swing motion towards a more elevated and lateral position to adjust the body's center-of-mass [7]. [7] noted that the magnitude and direction of such reactive arm-swing adaptations varied considerably across healthy participants despite a uniform perturbation being adminstered.

To ensure stability while also achieving adaptability, an optimal level of variability in the dynamics of gait is necessary [8]. This trade-off is a phenomenon that is present in various physiological signals (e.g. heart-rate variability) [9]. Variability above or below this optimal level can be indicative of pathology as excessive periodicity causes a system to be insufficiently adaptable to the demands of the environment while excessive variability introduces sub-optimal performance and instability [8,10]. Evidence from studies investigating this phenomenon through the dynamic systems perspective suggests that abnormal gait coordination variability is related to instability in patient populations [11,12]. Nonetheless among healthy adults, both high- and low-variability can be associated with gait stability indicating that other factors may also be at play in producing instable gait [13]. Variability and changes in variability however may also reflect the adaptability of gait and respective adaptions to environmental demands [14]. Therefore, the context and subject in question must be considered when assessing movement from this perspective.

Human movement is thought to be modularly controlled by the CNS through task-relevant synergistic muscle activations and in doing so, the CNS selects a near optimal solution to the motor-task from the many redundant degrees of freedom available [15,16]. Common, invariant muscle synergies have been identified across various activities such as standing and walking [17], while more task-specific synergies have also been revealed during slipping for instance [18], demonstrating the adaptability of human motor control to environmental demands [18]. revealed differences across healthy adults in terms of slip-severity were related to the magnitude of specific muscle activations in response to a perturbation (e.g. the medial hamstring at heel-strike post-slip). In a similar vein [19], demonstrated the different modular control the CNS exhibits on body segments in maintaining whole-body angular momentum during the swing-phase compared to the double-support phase of gait. Synergies have been successfully identified

in not just the muscular space but also in dynamic and kinematic spaces [20,21]. It must be noted however that the origin of these synergies and whether they represent an input or output of neuromuscular control is still a point of debate in the literature [16,22]. Nonetheless identifying deficits in adaptability across specific phases of gait from a dynamic systems perspective has the potential to inform targeted rehabilitation interventions and clinical assessments [23].

The measurement of discrete points in complex systems requires their synthesis to give more accurate and reliable interpretations. The *'curse of dimensionality'* is a problem encountered in this process as large feature sets are to be analysed simultaneously. Dimensionality reduction methods have become popular in this the era of *'big data'* as they alleviate this issue by reducing the number of features needed for a comprehensive analysis [24]. This reduction can be implemented via the creation of a smaller number of new variables that retain a large amount of the information in the original feature-set. Alternatively, through feature selection methods the most relevant feature-subset can be identified and extracted based on specific criteria. Dimensionality reduction has been used successfully to identify muscle synergies [25], contrast sporting techniques [23], improve fall-risk classification [26,27] etc. and their uses continue to expand with the availability of large datasets.

In particular and of specific relevance to this study, recent research has found effective use of principle component analysis (PCA) in reducing the dimensionality of kinematic data for various aims [21,28,29]. Of note, the '*Principle movements'* (PM) are the eigen-vectors derived from a PCA of movement data that represent correlated changes in marker coordinates (synergies) [21]. Each successive PM extracted from this analysis is ordered in terms of the amount of variance explained by the movement with higher-order PMs explaining less variance and representing more subtle movement patterns [30]. By projecting the $PM_k$ scores onto a posture space, the change in position of the body segments can be represented with respect to time and is known as the '*principal position'* ($PP_k$) [21]. In accordance with Newton's laws of motion, $PP_k$ time-series can be differentiated to '*principal velocities'* ($PV_k$) and '*principal accelerations'* ($PA_k$) representing the $1^{st}$- and $2^{nd}$-order derivatives of the $PP_k$ time-series [21]. As muscles activated by the nervous system causes changes in body segment motion, variables computed from $PA_k$ time-series have been of focus in related literature as the extracted acceleration signals are thought to represent the actions of the neuromuscular system on body segments [31–34].

In this study, a novel $PA_k$ variable ($PA_k$RMS) representing the magnitude of neuromuscular control on $PM_k$ will be derived from a gait analysis of healthy adults exposed to mechanical perturbations at various speeds. The aim of this study is to determine the utility of $PA_k$RMS in identifying key gait mechanisms that differentiate healthy individuals in terms of adaptability. To this end, the following associations will be investigated: 1) $PA_k$RMS and lower-limb coordination variability and 2) the moderating effect of $PA_k$RMS on the relationship between lower-limb coordination variability and smoothness of arm-swing motion. It is hypothesized for the first aim that $PA_k$RMS will demonstrate a negative effect on lower-limb coordination variability as participants who exhibit more control on specific $PM_k$ will require less adaptations and therefore exhibit lower coordination variability. For the secondary aim, it is hypothesized that $PA_k$RMS will also demonstrate a negative moderating effect on this upper-limb—lower-limb relationship as individuals who exhibit greater control of specific $PM_k$ will be able to prevent lower-limb perturbations from accumulating into trunk-level postural corrections.

## 2.1 Methods

### 2.1.1 Secondary data analysis

Thirty-nine trials of normal and perturbed treadmill walking collected from 13 participants were taken from an open-source dataset generated in the Human Motion and Control

Laboratory in Cleveland State University [35]. Characteristics of the included participants were: Age (Years) = 24±4.12, Height (Meters) = 1.74±0.08, Weight (kg) = 72.88±12. In short, participants were asked to walk normally on an R-Mill treadmill with dual 6-degrees of freedom force plates and independent belts for each foot (Forcelink, Culemborg, The Netherlands). Motion capture was undertaken using a 10 Ospreys camera motion capture system paired with the Cortex 3.1.1.1290 software at a sample rate of 100Hz (Motion Analysis, Santa Rosa, CA, USA). Participants were first asked to walk for 2 minutes unperturbed and were then subjected to 8 minutes of longitudinal belt-speed perturbations. This protocol was repeated for each of 0.8m/s, 1.2m/s and 1.6m/s walking speeds. The pseudo-random belt-speed control signals used to induce perturbations during each stance phase were generated a priori using MATLAB and Simulink (Mathworks, Natick, Massachusetts, USA) and are available online [36].

Marker trajectories from a 47 whole-body marker setup for both normal-walking and perturbed-walking trials were extracted and processed using the GaitAnalysis toolbox V 0.1.2 [37]. Data for normal-walking trials was taken from when a constant belt-speed was achieved (leaving approximately 50 seconds of normal-walking) and the length of the time-series were normalized to 5,000 data points and then low-pass filtered with a $4^{th}$-order Butterworth filter at a cut-off frequency of 20Hz.

### 2.1.2 Independent variable computation

All of the beforementioned procedures in this section were carried out using PManalyzer software [38], a Matlab GUI specifically designed for PM variable computation (Matlab (R2019B), Natick, Massachusetts: The MathWorks Inc.). Only data from normal-walking trials was included in the independent variable computation. In short and in line with other recent endeavours using this approach [32–34], the data from all subjects and walking-speed trials were pooled into one matrix to allow for direct comparisons between subjects and across trials. Raw marker coordinates (XYZ) were first transposed so that each frame represents a posture vector, then centred by subtracting each posture vector from the mean posture vector to get postural deviations and normalized to their mean Euclidean distances to ensure an equal contribution by each subject/trial to the pooled dataset and subject-specific overall variance respectively. The data was also centred towards the centre-of-mass to create a body-position dependent coordinate system thus removing the potential inclusion of irrelevant body displacements from the PMs [23]. The pooled dataset of concatenated normal-walking trials accumulated into a 195,000 x 60 input matrix (100 Hz [Sampling rate] x approximately 50 seconds [Trial duration] x 3 [Three trials] x 13 [Number of subjects] x 60 [Marker coordinates]). The PCA algorithm decomposed this input into a covariance matrix of $PM_k$.

The $PM_k$ within the output of this process were leave-one-out cross validated. The first 7 PMs presented a change of less than 15˚ when a participant was left out and were therefore included in further analysis [31,38]. To compute the $PA_k$ time-series, the $PP_k$ time-series were first extracted as the PCA-scores derived from the PM eigen-vectors that represent deviations in posture across the orthonormal posture-space (25). The time-series were then further low-pass filtered using a $3^{rd}$-order Butterworth filter at a cut-off of 10Hz and their spectral properties inspected with a Fourier analysis for frequency power above what is expected in noise free movement data [39]. No significant power was found at frequencies above 5Hz and so the effect of noise was deemed to have been sufficiently reduced. The control of $PM_k$ was represented as the root-mean-square (*RMS*) of the $PA_k$ time-series with respect to trial duration (*t*) (*$PA_kRMS$*) (Eq 1). In previous studies using this measure, static balance tasks were investigated where the accelerations in $PA_k$ components could be directly related to postural control

mechanisms [33]. During dynamic movements however, it must be noted that these accelerations additionally include dependencies such as walking-speed and so cannot solely reflect the actions of the neuromuscular control system [32]. Therefore, the $PA_kRMS$ were then normalized by the respective walking-speed of the trial.

$$PA_kRMS = \text{RMS}\left(\frac{PA_k}{t}\right) \qquad (1)$$

### 2.1.3 Dependent variable computation

Lower-limb coordination variability was represented as the stride-to-stride variability in inter-joint continuous relative phase (CRP) values known as the deviation phase (DP). Heel-strikes events during perturbed walking trials were identified using a coordinate-based treadmill algorithm [40]. CRP was calculated in accordance with the methodology described in [41] to determine the coordinative relationship between the Hip-Ankle, Hip-Knee and Knee-Ankle of both limbs in the sagittal plane only. Eq 3 below illustrates the CRP formula where $\varphi_{proximal}(t)$ is the phase normalized joint angle of the proximal segment at timepoint t and $\varphi_{distal}(t)$ is the phase normalized joint angle for the distal segment at the same timepoint. From this, $DP$ is calculated as the standard deviation of $CRP(t)$. The average DP across the CRP measurements within each perturbed walking-speed trial was then taken as a summary statistic for further analysis.

$$CRP(t) = \varphi_{proximal}(t) - \varphi_{distal}(t) \qquad (2)$$

$$DP = \sqrt{\frac{\sum (CRP(t) - CRP\mu)^2}{N}}$$

Arm-elevation jerk ($Jerk_v$) was computed as the 3rd-order derivative of the medial-wrist marker position trajectories ($\vec{r}(t)$) with respect to time (Eq 3). The mean absolute $Jerk_v$ ($|\overline{Jerk_v}|$) was then normalised by the peak velocity ($max(\frac{d\vec{r}(t)}{dt})$) of the marker trajectory to formulate a measure of movement smoothness adopted from [42] hereafter referred to as the normalised jerk index ($NJI$). As reactive arm-swing strategies are said to be asymmetrical (7), the side that exhibited the highest NJI value was included in statistical analyses.

$$Jerk_v = \frac{d^3\vec{r}(t)}{dt^3} \qquad (3)$$

$$NJI = \frac{|\overline{Jerk_v}|}{max(\frac{d\vec{r}(t)}{dt})}$$

### 2.1.3 Statistical analysis

Due to the repeated-measures structure of this study along with the hypotheses being tested, a linear mixed-effects regression was deemed appropriate. Mixed-effects regression is different to the traditional ordinary least-squares in that it can flexibly consider effects at different levels of analysis (i.e. multi-level regression), known as random-effects. Mixed-effects models are useful for data structures which may have clustering present, providing more accurate effect estimates than traditional techniques in such situations [43]. In the current study, clustering may occur as a result of multiple observations from the same participant while the extent to which the effect of $PA_kRMS$ varies across trials is also of interest to this study's aims [44]. suggested

that data-driven maximal models should be utilised in the case of confirmatory hypothesis modelling. Therefore, the following effect terms were included in the models for both aims:

**PA$_k$RMS**: Each PA$_k$RMS component will be input into separate models as fixed-effects, to examine how the magnitude of neuromuscular control in each movement synergy affects gait adaptability.

**Trial:** As gait variability is said to be speed-dependent [45], the fixed-effect of walking-speed was included in both models.

**PA$_k$RMS * Trial:** It is within the scope of this study's aims to examine the influence of walking-speed on the predictability of PA$_k$RMS on the dependent variables. Therefore, an interaction term was modelled in both aims.

**(PA$_k$RMS|Participants):** As a random sample of participants were taken from the population and dependency across observations is present due to repeated measures, subject-specific intercepts were modelled allowing within-participant clustering to be considered. This was accompanied by random slopes to allow the slope to vary across participants also.

**(PA$_k$RMSTrial):** As the walking-speeds in each trial were arbitrarily selected, the effect of this random selection on the data structure will be considered by the inclusion of a random trial intercepts term. Along with this random slopes for trials will also be included to allow for differences in the effect of PA$_k$RMS across trials to be considered.

Eqs 4 & 5 below illustrate the maximal models for each aim in R software syntax form. Due to the small sample size included in this analysis, a step-wise elimination of fixed and random-effect terms from these maximal models was conducted to improve parsimony. Each PA$_k$RMS was separately input into individual mixed-effects models with the above described terms and a step-wise elimination was carried out for each of the 7 models in both hypotheses. The output from this procedure was then compared in Anova with F-test p-values and Satterthwaite's degrees-of-freedom approximation to identify optimal models for both hypotheses. An optimal synergy was identified as it was likely that a number of synergies would be relevant to the dependent variables due to inherent interdependencies and reciprocal compensation rather than through a direct relationship [46,47]. To reduce familywise error accumulation from multiple comparisons, a Bonferroni correction was implemented in which a significance level of $p < 0.007$ was set a priori for the final models. These statistical procedures were conducted using the *'LmerTest'* package in R software [48]. Goodness-of-fit was assessed using Akaike information criterion (AIC), Bayesian's information criterion (BIC) and McFadden's R-squared while coefficient estimates and p-values were obtained using restricted maximum-likelihood and the Satterthwaite's degrees of freedom method respectively.

$$DP \sim PA_kRMS + PA_kRMS : Trial + Trial + (PA_kRMS|Trial) + (PA_kRMS|Participants) \quad (4)$$

$$DP \text{ x } NJI \sim PA_kRMS + PA_kRMS : Trial + Trial + (PA_kRMS|Trial) + (PA_kRMS|Participants) \quad (5)$$

## 3.1 Results

### 3.1.1 Walking condition variables

Average and standard deviation values for the variables of interest in this study are presented in Table 1 while a summary of each validated PM$_k$ is described in Fig 1. These graphical representations are also available as 2D and 3D videos in the supplementary material attached. DP consistently increased with walking-speed in the perturbed-walking trials from a low of 25.1

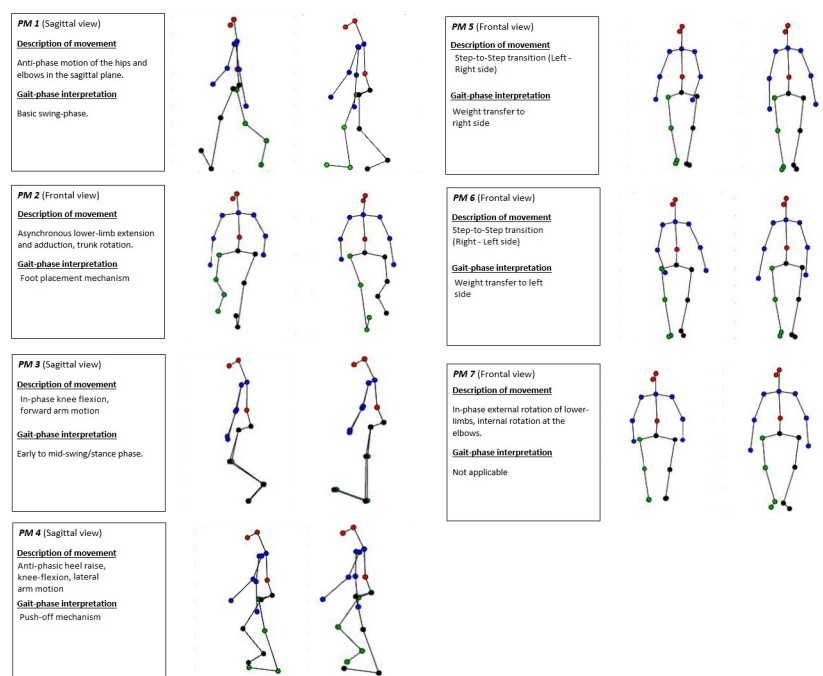

**Fig 1. Summaries of each of the validated $PM_k$ along with their respective graphical representations to the right.**
An amplification factor of 2 was used to provide clarity to more subtle movements in visualizations.

**Table 1. Mean and standard deviation for independent and dependent variables in both conditions and for each walking-speed.**

| Perturbed-walking condition | 0.8 m/s | 1.2 m/s | 1.6 m/s |
|---|---|---|---|
| DP | 25.1±3.5 | 27.8±2.5 | 28.6±2.4 |
| NJI | 0.16±0.07 | 0.22±0.14 | 0.24±0.14 |
| **Normal-walking condition** | **0.8 m/s** | **1.2 m/s** | **1.6 m/s** |
| $PA_1$ RMS* | 25.6±4.5 | 36.1±5.2 | 44.3±6.7 |
| $PA_2$ RMS* | 14.0±2.5 | 13.2±1.8 | 11.8±1.8 |
| $PA_3$ RMS* | 24.9±4.0 | 31.4±4.1 | 35.0±4.7 |
| $PA_4$ RMS* | 8.6±1.7 | 10.1±1.3 | 10.9±1.4 |
| $PA_5$RMS* | 4.1±1.0 | 4.8±1.1 | 5.6±1.4 |
| $PA_6$ RMS* | 5.0±1.0 | 5.3±0.9 | 5.8±1.1 |
| $PA_7$RMS* | 8.1±1.4 | 11.8±.17 | 14.6±2.4 |

(* non-normalized values for $PA_k$RMS).

±3.5 at 0.8m/s to a high of 28.6±2.4 at 1.6m/s. Of note however, the 0.8m/s trials experienced the highest between-participant variability in DP. The smoothness of arm-swing elevation during perturbed-walking trials (NJI) also increased with walking-speed from 0.16±0.07 at 0.8m/s to 0.24±0.14 at 1.6m/s. The non-normalized values for the first 7 validated $PA_k$RMS components taken from normal-walking trials are also presented. All $PA_k$RMS components demonstrated a positive relationship with walking-speed except in the case of $PA_2$RMS (representing the foot placement mechanism) which exhibited a negative relationship with walking-speed. The dominant movement captured using PCA was $PM_1$ (variance explained = 83.9%) that was

interpreted to represent a stiff inverted-pendulum motion of the lower-limbs along with elbow-flexion. $PM_3$ captured the synchronous extension of the knees during mid-swing/stance phase while $PM_4$ captured the push-off mechanism. $PM_5$ and $PM_6$ represent step-to-step transitions from the left—right side and right—left side respectively. $PM_7$ explained just 0.39% of the total variance and so a recognizable gait pattern could not be interpreted. It could be noted however that transverse plane rotations of the lower-limbs and elbows were present for $PM_7$. Fig 2 provides some insight into the output of the PCA computation in terms of cumulative percentage of variance explained by each $PM_k$ and their respective eigenvalues.

### 3.1.2 Identification of optimal models

Table 2 describes the parsimonious Eq 4 models found from a stepwise elimination of effect terms and the findings from a subsequent comparison of these models using Anova. Just two $PA_kRMS$ components were found to have individual relevance in explaining the variance in DP, that of $PA_2RMS$ and $PA_4RMS$. A random-participants intercepts term ((1 |Participant)) and a random-participants intercepts and slopes term ((PA_4RMS |Participant)) were also included in these Eq 4 models. The accompanying random-effect term to the $PA_4RMS$ model indicates that the effect of $PA_4RMS$ differs significantly in starting value and in the rate of change (i.e. slope) between-participants. The $PA_2RMS$ model was more parsimonious (degrees of freedom = 4) but demonstrated a slightly less goodness-of-fit than the $PA_4RMS$ model. This difference was not significant however, as indicated by the chi-square statistic ($p > 0.05$).

Table 3 describes the findings from a similar procedure with Eq 5 models. $PM_2RMS$– $PM_6RMS$ found individual relevance in moderating the relationship between lower-limb DP and the smoothness of arm-swing elevation(DP x NJI). All of these models were accompanied by a random-participants intercepts term ((1 |Participant)), indicating within-participant clustering was a predominant level II effect. The $PM_2RMS$ Eq 5 model was used as the baseline model for the Anova. Three Eq 5 models were found to be significantly different from this baseline model in terms of goodness-of-fit, with $PM_6RMS$ conveying the best-fit.

### 3.1.3 Final mixed-effects models

Table 4 details the regression output from the final, optimal models for both Eqs 4 & 5 found from the stepwise elimination procedure. Both of the models fixed-effects reached a Bonferroni corrected significance level of $p < 0.007$. $PA_4RMS$, the fixed-effect term in the Eq 4 model, had a negative effect on DP ($\beta = -2.18 \pm 0.51$) and explained 37% of the variance in DP (McFadden's Pseudo $R^2 = 0.37$). With the addition of the random-participants intercepts and slopes term, the Eq 4 model explained 70% of the total variance in DP (McFadden's

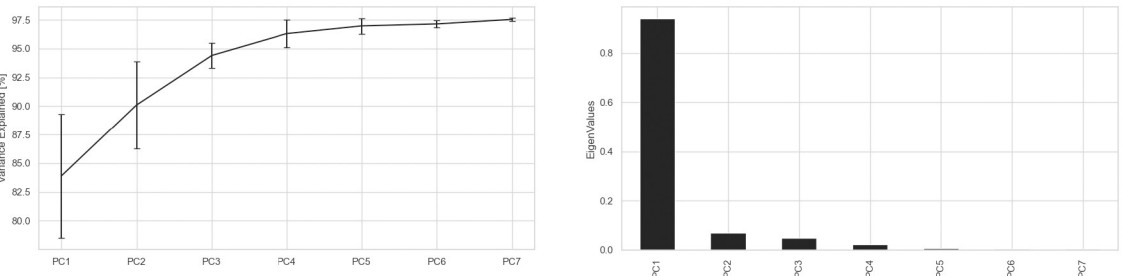

**Fig 2. Illustrations of both the percentage of variance explained (left) and Eigenvalues (right) for each $PM_k$.**

**Table 2. Output from a comparison of identified Eq 4 models using Anova.** The PA$_4$RMS model was found to be the best fit however insignificantly.

| Goodness-of-Fit indices | DP ~ PA$_2$RMS + (1 |Participant) | DP ~ PA$_4$RMS + (PA$_4$RMS |Participant) |
|---|---|---|
| Degrees of freedom | 4 | 6 |
| AIC | 192.5 | 191.0 |
| BIC | 199.2 | 201.0 |
| Log-likelihood | -92.3 | -89.5 |
| Deviance | 184.5 | 179.0 |
| Chi-square (p-value) | - | 5.5 (p>0.05) |

**Table 3. Output from a comparison of identified Eq 5 models using Anova.** The PA$_6$RMS model was found to be the best fit.

| Goodness-of-Fit indices | DP x *NJI* ~ *PA*$_2$RMS +(1|*Participants*) | DP x *NJI* ~ *PA*$_3$RMS +(1|*Participants*) | DP x *NJI* ~ *PA*$_4$RMS +(1|*Participants*) | DP x *NJI* ~ *PA*$_5$RMS +(1|*Participants*) | DP x *NJI* ~ *PA*$_6$RMS +(1|*Participants*) |
|---|---|---|---|---|---|
| Degrees of freedom | 4 | 4 | 4 | 4 | 4 |
| AIC | 207.2 | 207.9 | 207.1 | 206.7 | 205.0 |
| BIC | 213.8 | 214.5 | 213.7 | 213.4 | 211.6 |
| Log-likelihood | -99.6 | -99.9 | -99.5 | -99.4 | -98.5 |
| Deviance | 199.2 | 199.9 | 199.1 | 198.7 | 197.0 |
| Chi-square (p-value) | - | 0.00 (p = 1.00) | 0.8 (p<0.0001) | 0.4 (p<0.0001) | 1.8 (p<0.0001) |

**Table 4. Optimal models found for Eqs 4 & 5.**

| Regression output | Eq 4 DP ~ PA$_4$RMS + (PA$_4$RMS | Participant) | Eq 5 DP x NJI ~ PA$_6$RMS + (1| Participant) |
|---|---|---|
| FE β Coefficient * | -2.18 | -1.02 |
| FE Standard error* | 0.51 | 0.33 |
| P-value | <0.005 | <0.005 |
| RE-Participant intercepts variance* | 3.08 | 2.27 |
| RE-Participant slopes variance* | 1.37 | - |
| RE-residual variance* | 3.84 | 5.32 |
| Inter-class coefficient (ICC) | 0.44 | 0.34 |
| Pseudo-R$^2$ (Fixed-effects only) | 0.37 | 0.17 |
| Pseudo-R$^2$ (Total) | 0.70 | 0.45 |
| Restricted-Maximum Likelihood | 178.3 | 196.3 |

(*FE = Fixed effect, RE = Random effect).

Pseudo R$^2$ = 0.37). The variance within the data that could be explained by within-participant clustering was $\sigma^2$ = 3.08 in the Eq 4 model and to a lesser extent in the Eq 5 model ($\sigma^2$ = 2.27). Between-cluster variance was also predominant in the data as the random-effect residual variance was $\sigma^2$ = 3.84 in the Eq 4 model and $\sigma^2$ = 5.32 in the Eq 5 model. The inter-class correlation within both models was moderate with an ICC of 0.44 and 0.34 for Eqs 4 & 5 respectively, indicating the presence of clustering in the data and further demonstrating the

necessity for a mixed-effects regression approach. Random-slopes for participants were also included in the Eq 4 model and this term captured 1.37 of the variances.

## 4.1 Discussion

This study sought to investigate the role of the neuromuscular system in controlling key movement synergies and their capacity to differentiate healthy adults in terms of gait adaptability across various speeds. For this investigation, a coordination metric was devised in the form of deviation phase (DP) that would indicate the level of gait adaptations required in the lower-limbs during a trial of perturbed-walking on a treadmill. The association between this metric and PCA derived measures ($PA_kRMS$) representing the magnitude of neuromuscular control on normal-walking movement synergies was then determined using linear mixed-effects models. The hypothesis that specific $PA_kRMS$ would have a negative effect on DP was substantiated. As a secondary aim that extended this finding, the current study investigated the moderating effect of $PA_kRMS$ on the relationship between DP and the smoothness of arm-swing elevation (DP x NJI) during perturbed-walking. The hypothesis that $PA_kRMS$ would demonstrate a negative effect on this relationship was also substantiated. A stepwise approach was undertaken for both aims as it was foreseeable that a number of movement synergies could be related to DP and (DP x NJI) due to reciprocal compensation [47]. The identification of an optimal model allowed for the main source for these differences to be determined and demonstrating the usability of such a protocol for informing targeted rehabilitation and clinical assessments. Interestingly, a number of the identified synergies have been described in relevant research including the push-off and foot placement mechanisms [5,46]. The current study found the push-off mechanism to be optimal in explaining variance in DP while synergies related to the transfer of weight to the left side were also most relevant for the DP x NJI interaction.

### 4.1.1 Aim 1

The push-off mechanism has been observed in previous research where the modulation of ankle plantar-flexion during a perturbation allowed for the modulation of other gait parameters (i.e. step length and width) [5,46]. Participants in the current study exhibiting a greater control during push-off prior to the slip-event were perhaps more capable of recovering posture by avoiding an excessive foot-floor angle post-slip, a gait parameter associated with slip severity [49,50]. found significant associations between slip severity and baseline muscle synergy patterns among healthy adults. Less severe slippers demonstrated higher activation of the medial-hamstring at heel-strike and of the vastus-lateralis following heel-strike allowing for more efficient weight transfer than their less stable counterparts. The findings of the current study mirror this in that the foot placement mechanism was related to DP during a trial of perturbed-walking [50]. focused solely on post-slip synergies and so understandably, the proactive role of the push-off mechanism for gait adaptations was not elicited.

[19] noted the closed-loop control of gait during the double-support phase and how the CNS appears to use this phase to correct whole-body angular momentum while swing-phase plays a reactive, stabilising role in terms of stride-to-stride reproducibility. It is therefore unsurprising to find a movement synergy identified during the double-support phase of normal-walking was most relevant to DP as this gait-phase plays a proactive role in dynamic balance control. Interestingly [51], was unable to find differences between mild- and severe-slippers in terms of baseline sagittal plane angular momentum, centre-of-mass height, single/double-support duration or upper-body extremity kinematics but did find differences between groups during perturbed-walking. The results of the current study suggest PCA derived

movement synergies may be useful in identifying such differences relevant to slip severity during normal-walking trials. Due to the significant random-participant slopes term within the Eq 4 model, the current study's results also suggest that the closed-loop control represented in $PA_4RMS$ may differ in its effect on gait adaptability significantly across walking-speeds. As this random-effects term was participant-specific rather than trial-specific, this relationship may potentially manifest as a function of the participants preferred walking-speed [45]. Further research is required however to substantiate this claim.

### 4.1.2 Aim 2

Three $PA_kRMS$ were identified to contribute a moderating effect, that of $PA_4RMS$, $PA_5RMS$ and $PA_6RMS$ which were interpreted to represent the push-off mechanism and weight-transfer to the right and left sides respectively. Moreover, $PA_6RMS$ was identified to be the most optimal synergy in explaining the variance in DP x NJI, indicating the central role of this synergy in differentiating healthy participants [7]. established the role of the upper-limbs in adjusting the centre-of-mass position when a perturbation is experienced. These arm-motion strategies were said to include increased elevation and lateral orientation and depended on the side of the perturbation experienced. They found the frontal plane position of the trunk at heel-strike to be related to perturbed-side arm elevation in that with increased trunk lean towards the trailing limb-side, the CNS intervened by adjusting the elevation of the perturbed-side arm to improve the centre-of-mass position. The current study would support these findings with relevant movement synergies identified during the double-support and weight-transfer phases of gait.

In patient populations where reduced gait stability is present (e.g. stroke), the transfer of weight is typically difficult as strength deficits among extensor muscles make this transfer less efficient [52]. Within healthy populations however, where muscle strength should be sufficient to support normal gait, more subtle differences may still be present that become relevant when compared across healthy participants. Promsri and colleagues investigated the effect of leg dominance on balance control during over-ground single leg-stance [53] and also single-legged stance on a multiaxial unstable board [39]. Sufficient evidence was found to warrant the consideration of leg dominance in clinical testing and balance training. In the current study, it is interesting that $PA_6RMS$, representing right-to-left side weight transfer, was the most relevant movement synergy in this sample despite explaining less variance than the opposite side ($PA_5RMS$). Assuming this random sample of healthy adults who were invited to take part in this study were in fact randomly recruited, then it is reasonable to propose that the majority of participants were right-side dominant [54,55]. Combining this insight with that of the aforementioned observations made by [7], we propose that asymmetry in limb control between the dominant and non-dominant sides was relevant in differentiating healthy adults in terms of gait adaptability in this study. The moderating effect of movement synergies on coupled asymmetries between the upper- and lower-limbs has been found in recent research and may have significant influence on gait performance and risk for injury [56]. The influence of these inherent asymmetries on gait stability has not been investigated in the research thus far. We therefore concur with the aforementioned studies related to leg dominance that this phenomenon should be considered in balance testing and clinical evaluations.

## 5.1 Limitations

Healthy adults in this study were subjected to perturbations during treadmill walking and therefore the findings cannot be generalized directly to over-ground walking. Results should be interpreted with the limitations of a relatively small sample size in mind.

## 6.1 Conclusion

This study identified movement synergies sourced from normal-walking trials whose magnitude of neuromuscular control had significant negative effects on lower-limb coordination variability during perturbed-walking at various walking-speeds. Relevant movement synergies were also identified that moderated the relationship between lower-limb coordination variability and the smoothness of arm-swing motions, thus indicating the utility of this approach in differentiating healthy adults in terms of gait adaptability. The usability of this approach in informing targeted rehabilitation and clinical assessment was supported. Future research should investigate further the change in control of these movement synergies across walking-speeds, the influence of leg dominance on gait adaptability and the utility of this analytical approach in patient populations.

## Supporting information

**S1 Data. The data used in this study is publicly available and peer reviewed [35].** Both 2D and 3D graphical representations of all validated $PM_k$ displayed in Fig 1 of this manuscript are available in video format at the following Github repository: https://github.com/Davidoreilly12/O-Reilly-Federolf-2020.
(DOCX)

## Author Contributions

**Conceptualization:** David Ó'Reilly.

**Formal analysis:** David Ó'Reilly.

**Investigation:** David Ó'Reilly.

**Methodology:** David Ó'Reilly.

**Resources:** Peter Federolf.

**Software:** David Ó'Reilly.

**Supervision:** Peter Federolf.

**Visualization:** David Ó'Reilly.

**Writing – original draft:** David Ó'Reilly.

**Writing – review & editing:** Peter Federolf.

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
