## [Decision Letter · Decision Letter 0]

23 Oct 2020

PONE-D-20-23630

Identifying differences in gait adaptability across various speeds using movement synergy analysis

PLOS ONE

Dear Dr. Oreilly,

Thank you for submitting your manuscript to PLOS ONE. After careful consideration, we feel that it has merit but does not fully meet PLOS ONE’s publication criteria as it currently stands. Therefore, we invite you to submit a revised version of the manuscript that addresses the points raised during the review process.

Dear authors,

The manuscript had positive revisions. However, minor revisions are necessary yet. Please address them.

We look forward to receiving your revised manuscript.

Kind regards,

Fabio A. Barbieri, PhD

Academic Editor

PLOS ONE

Journal Requirements:

2. Please amend your list of authors on the manuscript to ensure that each author is linked to an affiliation. Authors’ affiliations should reflect the institution where the work was done (if authors moved subsequently, you can also list the new affiliation stating “current affiliation:….” as necessary).

Reviewers' comments:

Reviewer's Responses to Questions

**Comments to the Author**

1. Is the manuscript technically sound, and do the data support the conclusions?

Reviewer #1: Yes

Reviewer #2: Partly

2. Has the statistical analysis been performed appropriately and rigorously? 

Reviewer #1: Yes

Reviewer #2: Yes

3. Have the authors made all data underlying the findings in their manuscript fully available?

Reviewer #1: Yes

Reviewer #2: Yes

4. Is the manuscript presented in an intelligible fashion and written in standard English?

Reviewer #1: Yes

Reviewer #2: Yes

5. Review Comments to the Author

Reviewer #1: This study evaluates a novel approach to represent the size of neuromuscular control the PAkRMS with respect to co-ordination variability and arm swing. hypothesis clearly stated.

Detailed description of what was done.

clearly presented results

Good discussion, well done very interesting and relevant.

Reviewer #2: Dear Mr. O’Reilly,

I thoroughly enjoyed reading your manuscript titled, “Identifying differences in gait adaptability across various speeds using movement synergy analysis.” I believe that it presents a helpful contribution to the fields of gait analysis and postural adaptability, and it fits well within the scope and publication criteria of this journal. I also found it to be very well written, I appreciate your style and your ability to speak both generally and specifically within this computationally intense method. I do recommend some minor revisions, some of which are merely text-level revisions, which I will detail by section below.

Abstract

-There are many abbreviations in the abstract, some of which could be minimized. PA4RMS and PA6RMS, for example, would have little meaning to someone who reads only the abstract, and they confused me on first read. I would recommend writing in more general terms, e.g., “The principal movement component relating to push-off was identified…” (line 29).

Introduction

-Line 89, bodys center of mass is missing an apostrophe.

-Line 109 says “specific phases gait…,” it appears to be missing a preposition.

Methods

-Line 196, Equation 2. Your outcome of interest here is deviation phase, not CRP. I would recommend using Equation 2 to define DP. You can include CRP within your definition or do a two-line definition like you did with NJI, e.g.,

DP=std(CRP(t))=std(φ_proximal (t)- φ_distal (t))

-In the statistical analysis section I think you do an excellent job providing rationale for the regression model that you selected.

-Lines 217-218, overall, I enjoy the innovative method of dimensionality reduction in kinematic data to look at whole-body adaptation strategies within these principal movements. I think it does a good job summarizing the high-dimensional data that biomechanists tend to distill down into a joint-by-joint and plane-by-plane analysis of point data. However, I wonder if it may be too soon to connect a derived metric like PAkRMS with the concept of neuromuscular control. It is a strong topic of debate whether the synergies derived from EMG in muscle synergy analysis represent any true aspect of neuromuscular control (1,2). Kinematics are a step further removed from this discussion, because you there are no neuromuscular signals being measured. This could be indicative of behavior, and perhaps with further research it could be validated as an indirect measure of neuromuscular control. Nevertheless, I believe it may be too soon to draw the connection at the moment.

Results

-Line 260 says mid-sing rather than mid-swing.

-Figure 1/Section 3.1.1. I have a few questions regarding the extracted principal movements:

-Should the variance explained total to 100%? The first 2 PM’s alone add to 101.1% and PMs 1-7 total 109.74%.

-PM’s 5-7 each account for <1% of the variance. Other dimensionality reduction techniques typically use a cutoff or other criterion for the number of principal components that likely contribute meaningfully to the patterns of behavior observed. One could argue that retaining and drawing conclusions from the highest order components would be akin to fitting noise. The results go on to implement these PMs into statistical models and even identify PA6RMS as the best fit model of the DP x NJI relationship. Has the use of this method to explain direct contributions of the highest order components to movement patterns been validated? Do the eigenvectors provide insights about whether these components should remain?

-In Figure 1’s caption, it is noted that “an amplification factor of 2 was used to provide clarity to more subtle movements.” How did you go about deciding which movements to amplify? Is it possible that, per the comments above, movements that were too subtle to detect without amplification might not warrant inclusion within these results?

-In Table 4, it would be more consistent with the rest of the manuscript if Equation 5 said DP x NJI, rather than DP x Jerk.

Discussion

-No comments.

Works Cited in Methods Comment

1. Alessandro C, Delis I, Nori F, Panzeri S, Berret B. Muscle synergies in neuroscience and robotics: from input-space to task-space perspectives. Front Comput Neurosci [Internet]. 2013 [cited 2020 Oct 19];7. Available from: http://journal.frontiersin.org/article/10.3389/fncom.2013.00043/abstract

2. Zelik KE, La Scaleia V, Ivanenko YP, Lacquaniti F. Can modular strategies simplify neural control of multidirectional human locomotion? J Neurophysiol. 2014 Jan 1. 2014;111:1686–702.

6. PLOS authors have the option to publish the peer review history of their article (what does this mean?). If published, this will include your full peer review and any attached files.

Reviewer #1: No

Reviewer #2: **Yes: **Caitlin L Banks

---

## [Author Response · Author response to Decision Letter 0]

26 Oct 2020

Dear Editor,

 I am writing to you in response to the recent revision given to the manuscript “Identifying differences in gait adaptability using movement synergy analysis”. Firstly, I would like to thank you for the timely response and the service you provide. As advised, consideration has been made further to formatting this manuscript towards the requirements of the Plos One journal. These formatting changes include headings font, author affiliations, reference style and figures among others.

The following is the authors response to both reviewers:

Reviewer 1

Thank you for taking the time to review this manuscript. I appreciate the kind words regarding the writing style. This is indeed encouraging for future work. 

Reviewer 2

From the outset, thank you for taking the time to review this manuscript and we’re glad you enjoyed reading it. Both authors have read the points raised and find them valid and also find that they will ultimately improve the manuscript. You can find a response to each point you have made below:

“Abstract

-There are many abbreviations in the abstract, some of which could be minimized. PA4RMS and PA6RMS, for example, would have little meaning to someone who reads only the abstract, and they confused me on first read. I would recommend writing in more general terms, e.g., “The principal movement component relating to push-off was identified…” (line 29).”

The authors agree with this and have amended the abstract by adding a quick description of both PA4RMS and P6RMS as the push off mechanism and Right – Left side weight transfer respectively, thus offering a more interpretable quick read. 

“Introduction

-Line 89, bodys center of mass is missing an apostrophe.

-Line 109 says “specific phases gait…,” it appears to be missing a preposition.”

The authors agree with this and thank the reviewer for their consideration. Both lines have been amended accordingly in the new manuscript version.

“Methods

-Line 196, Equation 2. Your outcome of interest here is deviation phase, not CRP. I would recommend using Equation 2 to define DP. You can include CRP within your definition or do a two-line definition like you did with NJI, e.g.,

DP=std(CRP(t))=std(φ_proximal (t)- φ_distal (t))

-In the statistical analysis section I think you do an excellent job providing rationale for the regression model that you selected.”

The authors agree with the above point made. The calculation of Deviation phase has now been presented in the manuscript as advised.

“-Lines 217-218, overall, I enjoy the innovative method of dimensionality reduction in kinematic data to look at whole-body adaptation strategies within these principal movements. I think it does a good job summarizing the high-dimensional data that biomechanists tend to distill down into a joint-by-joint and plane-by-plane analysis of point data. However, I wonder if it may be too soon to connect a derived metric like PAkRMS with the concept of neuromuscular control. It is a strong topic of debate whether the synergies derived from EMG in muscle synergy analysis represent any true aspect of neuromuscular control (1,2). Kinematics are a step further removed from this discussion, because you there are no neuromuscular signals being measured. This could be indicative of behavior, and perhaps with further research it could be validated as an indirect measure of neuromuscular control. Nevertheless, I believe it may be too soon to draw the connection at the moment.”

Thank you for this comment. The authors understand the concern raised but have chosen to continue with the use of neuromuscular control in the revised version. This decision was made based on the following considerations: 

- Kinematic synergies in this manuscript merely represent correlated changes in marker positions (as specified in line 119) and therefore do not infer the origin of neuromuscular control. The PAk variable captures accelerations in the marker coordinates that behave in a synergistic fashion. As muscles which are activated by the nervous system pull body segments to cause movement it is reasonable to describe these variables in terms of neuromuscular control. Such synergies are similar to that of M-modes in the literature (Krishnamoorthy et al., 2003). The following article goes into detail on the PAk variable and what it captures while also using this terminology (Longo et al., 2019). 

- The authors believe that the paragraph (line 97 – 108) may cause some confusion with readers in regards to what the PAk variable captures and therefore have added the following statement, “Synergies have been successfully identified in not just the muscular space but also in dynamic and kinematic spaces (Federolf, 2016; Grinyagin et al., 2005). It must be noted however that the origin of these synergies and whether they represent an input or output of neuromuscular control is still a point of debate in the literature (Bizzi and Cheung, 2013; Tresch and Jarc, 2009). Nonetheless identifying deficits in adaptability across specific phases of gait from a dynamic systems perspective has the potential to inform targeted rehabilitation interventions and clinical assessments (Federolf et al., 2014).”… We hope this encapsulates the current position of the research on this topic while the explanation of what a kinematic synergy is at line 119 ensures the reader understands that it is a reflection of synergistic behaviour rather than an inference of neural origin.

- The authors have also attempted further clarification in the introduction with the following addition from line 124 – 126: “As muscles activated by the nervous system causes changes in body segment motion, variables computed from PAk time-series have been of focus in related literature as the extracted acceleration signals represent the actions of the neuromuscular system on body segments (Haid et al., 2018; Longo et al., 2019; Promsri et al., 2020; Zago et al., 2019).”

“Results

-Line 260 says mid-sing rather than mid-swing.”

The authors have made adjustments accordingly.

“-Figure 1/Section 3.1.1. I have a few questions regarding the extracted principal movements:

-Should the variance explained total to 100%? The first 2 PM’s alone add to 101.1% and PMs 1-7 total 109.74%.”

This figure has been adjusted with the correct variance explained. The 7 principal components now sum to 97.8% of total variance.

“-PM’s 5-7 each account for <1% of the variance. Other dimensionality reduction techniques typically use a cutoff or other criterion for the number of principal components that likely contribute meaningfully to the patterns of behavior observed. One could argue that retaining and drawing conclusions from the highest order components would be akin to fitting noise. The results go on to implement these PMs into statistical models and even identify PA6RMS as the best fit model of the DP x NJI relationship. Has the use of this method to explain direct contributions of the highest order components to movement patterns been validated? Do the eigenvectors provide insights about whether these components should remain?” 

This is a very valid remark made by the reviewer. The authors however would like to point the reviewer to lines 170 – 171 where a cross-validation on the principal components derived is detailed and a cut-off of 15 degrees is described. From this cross-validation, the first 7 components did not significantly change when a subject was left out of the analysis, indicating that the components although capturing a small % of variance are robust across subjects. This cross-validation protocol has been carried out in the following paper also (Haid et al., 2018).

“-In Figure 1’s caption, it is noted that “an amplification factor of 2 was used to provide clarity to more subtle movements.” How did you go about deciding which movements to amplify? Is it possible that, per the comments above, movements that were too subtle to detect without amplification might not warrant inclusion within these results?”

The use of an amplification factor in this study was simply for interpretation purposes during analysis and visualizations in the graph. No amplification factor was implemented on variables used in the statistical analysis. For a brief on how this amplification factor is implemented please see the following references (Federolf, 2016; Haid et al., 2019). The authors recognize the concern raised regarding the use of higher order principal components however the above responses should address this.

“-In Table 4, it would be more consistent with the rest of the manuscript if Equation 5 said DP x NJI, rather than DP x Jerk.”

The authors have made the terminology more consistent across the manuscript and thank the reviewer.

References 

Federolf, P.A., 2016. A novel approach to study human posture control: “Principal movements” obtained from a principal component analysis of kinematic marker data. Journal of Biomechanics 49, 364–370. https://doi.org/10.1016/j.jbiomech.2015.12.030

Haid, T.H., Doix, A.-C.M., Nigg, B.M., Federolf, P.A., 2018. Age effects in postural control analyzed via a principal component analysis of kinematic data and interpreted in relation to predictions of the optimal feedback control theory. Frontiers in aging neuroscience 10, 22.

Haid, T.H., Zago, M., Promsri, A., Doix, A.-C.M., Federolf, P.A., 2019. PManalyzer: A software facilitating the study of sensorimotor control of whole-body movements. Frontiers in neuroinformatics 13, 24.

Krishnamoorthy, V., Latash, M.L., Scholz, J.P., Zatsiorsky, V.M., 2003. Muscle synergies during shifts of the center of pressure by standing persons. Experimental Brain Research 152, 281–292. https://doi.org/10.1007/s00221-003-1574-6

Longo, A., Haid, T., Meulenbroek, R., Federolf, P., 2019. Biomechanics in posture space: Properties and relevance of principal accelerations for characterizing movement control. Journal of Biomechanics 82, 397–403.

---

## [Decision Letter · Decision Letter 1]

20 Nov 2020

PONE-D-20-23630R1

Identifying differences in gait adaptability across various speeds using movement synergy analysis

PLOS ONE

Dear Dr. Oreilly,

Thank you for submitting your manuscript to PLOS ONE. After careful consideration, we feel that it has merit but does not fully meet PLOS ONE’s publication criteria as it currently stands. Therefore, we invite you to submit a revised version of the manuscript that addresses the points raised during the review process.

Dear authors,

One reviewer has indicated minor revisions for your manuscript. The reviewer indicated two specific points that I suggest that you addressed adequately, please.

We look forward to receiving your revised manuscript.

Kind regards,

Fabio A. Barbieri, PhD

Academic Editor

PLOS ONE

Reviewers' comments:

Reviewer's Responses to Questions

**Comments to the Author**

1. If the authors have adequately addressed your comments raised in a previous round of review and you feel that this manuscript is now acceptable for publication, you may indicate that here to bypass the “Comments to the Author” section, enter your conflict of interest statement in the “Confidential to Editor” section, and submit your "Accept" recommendation.

Reviewer #2: (No Response)

2. Is the manuscript technically sound, and do the data support the conclusions?

Reviewer #2: Yes

3. Has the statistical analysis been performed appropriately and rigorously? 

Reviewer #2: Yes

4. Have the authors made all data underlying the findings in their manuscript fully available?

Reviewer #2: Yes

5. Is the manuscript presented in an intelligible fashion and written in standard English?

Reviewer #2: Yes

6. Review Comments to the Author

Reviewer #2: I appreciate that the authors provided a timely resubmission which acknowledged each comment from the first round of review. There are two points that I would like to discuss further.

In lines 119-123 of the introduction, the authors have modified some content to clarify the work that has been done that leads them to conclude the PAk time series represent actions of the neuromuscular system. After reviewing the string of self-citations provided in the manuscript body and the response to reviewers, I cannot conclude whether the field would either agree or disagree with the characterization of PAk as representative of neuromuscular control. I do not wish to stand in the way of good work being published due to a disagreement on the language used to describe a metric. However, it would be my preference that the authors would blend the first and second submission versions of line 121, to keep the more reserved tone and state “…as the extracted acceleration signals are thought to represent the actions of the neuromuscular system on body segments,” as this more adequately reflects the stance of the authors in the comments to the reviewers.

Regarding the robustness of components across subjects in the methods section lines 163-164, I will acknowledge that I am not used to seeing a cross-validation of this type be the deciding factor in how many principal components get retained. I do still have reservations about maintaining and later drawing conclusions about the PMs that account for such a small proportion of the variance. If the authors are willing to provide the eigenvalues of the PMs displayed in Figure 1, this would be in keeping with what is done in previous studies using this technique (e.g., Haid et al. 2018, Longo et al. 2019). This way the reader will be provided with an additional piece of information that allows them to judge the contribution of these PMs for themselves.

Ultimately, I do conclude that this work is worthy of publication in PLOS One. I have no issues with the remainder of the manuscript. If the authors are willing to provide the eigenvalues noted above then I am willing to accept this manuscript for publication.

7. PLOS authors have the option to publish the peer review history of their article (what does this mean?). If published, this will include your full peer review and any attached files.

Reviewer #2: **Yes: **Caitlin Banks

---

## [Author Response · Author response to Decision Letter 1]

20 Nov 2020

Dear Editor,

I am writing in response to the second minor revision of the manuscript entitled “Identifying differences in gait adaptability across various speeds using movement synergy analysis”. Thank you for the timely turn around in revision. I have read the reviewers comments and you may find the responses to both below:

Reviewer 2

Thank you for your careful consideration of this manuscript and for the quick turnaround in reviewing it. Having read the comments made, you can find a response to both in the following:

“In lines 119-123 of the introduction, the authors have modified some content to clarify the work that has been done that leads them to conclude the PAk time series represent actions of the neuromuscular system. After reviewing the string of self-citations provided in the manuscript body and the response to reviewers, I cannot conclude whether the field would either agree or disagree with the characterization of PAk as representative of neuromuscular control. I do not wish to stand in the way of good work being published due to a disagreement on the language used to describe a metric. However, it would be my preference that the authors would blend the first and second submission versions of line 121, to keep the more reserved tone and state “…as the extracted acceleration signals are thought to represent the actions of the neuromuscular system on body segments,” as this more adequately reflects the stance of the authors in the comments to the reviewers.”

In the previously submitted manuscript, the authors made attempts to clarify the definition of the PAk variable noting the current position of the research on muscle synergies and providing a clear description of what this variable is capturing. The authors have also pointed towards previous research using this variable and so wish to stay in line with these studies and their definition used. The authors agree with the amendment recommended by the reviewer and have changed line 121 to the following: “…variables computed from PAk time-series have been of focus in related literature as the extracted acceleration signals are thought to represent the actions of the neuromuscular system on body segments…”. We hope this conveys a more conservative view on the variable to the reader and we again thank the reviewer for their considerations.

“Regarding the robustness of components across subjects in the methods section lines 163-164, I will acknowledge that I am not used to seeing a cross-validation of this type be the deciding factor in how many principal components get retained. I do still have reservations about maintaining and later drawing conclusions about the PMs that account for such a small proportion of the variance. If the authors are willing to provide the eigenvalues of the PMs displayed in Figure 1, this would be in keeping with what is done in previous studies using this technique (e.g., Haid et al. 2018, Longo et al. 2019). This way the reader will be provided with an additional piece of information that allows them to judge the contribution of these PMs for themselves.”

The authors agree that eigenvalues should be included. We have therefore created another figure (Fig 2.) directly below Figure 1 that illustrates both the percentage of variance explained by each component and the eigenvalues. The reader can thus see that although a cross validation of these components revealed that these components are adequately stable, some indeed explain a very small amount of the variance in the data.

---

## [Decision Letter · Decision Letter 2]

14 Dec 2020

Identifying differences in gait adaptability across various speeds using movement synergy analysis

PONE-D-20-23630R2

Dear Dr. Oreilly,

We’re pleased to inform you that your manuscript has been judged scientifically suitable for publication and will be formally accepted for publication once it meets all outstanding technical requirements.

Kind regards,

Fabio A. Barbieri, PhD

Academic Editor

PLOS ONE

Additional Editor Comments (optional):

Reviewers' comments:

Reviewer's Responses to Questions

**Comments to the Author**

1. If the authors have adequately addressed your comments raised in a previous round of review and you feel that this manuscript is now acceptable for publication, you may indicate that here to bypass the “Comments to the Author” section, enter your conflict of interest statement in the “Confidential to Editor” section, and submit your "Accept" recommendation.

Reviewer #2: All comments have been addressed

2. Is the manuscript technically sound, and do the data support the conclusions?

Reviewer #2: (No Response)

3. Has the statistical analysis been performed appropriately and rigorously? 

Reviewer #2: (No Response)

4. Have the authors made all data underlying the findings in their manuscript fully available?

Reviewer #2: (No Response)

5. Is the manuscript presented in an intelligible fashion and written in standard English?

Reviewer #2: (No Response)

6. Review Comments to the Author

Reviewer #2: All comments have been appropriately addressed and I recommend that this manuscript is accepted for publication.

7. PLOS authors have the option to publish the peer review history of their article (what does this mean?). If published, this will include your full peer review and any attached files.

Reviewer #2: **Yes: **Caitlin Banks

---

## [Editor Report · Acceptance letter]

16 Dec 2020

PONE-D-20-23630R2 

Identifying differences in gait adaptability across various speeds using movement synergy analysis 

Dear Dr. Ó' Reilly:

I'm pleased to inform you that your manuscript has been deemed suitable for publication in PLOS ONE. Congratulations! Your manuscript is now with our production department. 

Kind regards, 

on behalf of

Dr. Fabio A. Barbieri 

Academic Editor

PLOS ONE